# Quiescence Multiverse

**DOI:** 10.3390/biom15070960

**Published:** 2025-07-04

**Authors:** Damien Laporte, Isabelle Sagot

**Affiliations:** CNRS, Institut de Biochimie et Génétique Cellulaires, UMR5095, Université de Bordeaux, 33077 Bordeaux, France

**Keywords:** ageing, quiescence, stem cells

## Abstract

Cellular quiescence is operationally defined as a temporary and reversible cessation of proliferation. This state encompasses a wide range of physiological situations since most cells, from microbes to cells composing complex tissues, spend most of their lives non-dividing, waiting for signals to reproliferate. As such, individual quiescent cells must withstand the effects of time not only to survive but also to maintain their ability to divide. These capacities are shaped by a combination of deterministic factors relying on cell history and cumulative stochastic events linked to the environment but also to time. In addition, with time, quiescence deepens, the quiescence exit process being extended. Yet, this deepening is not necessarily sensed evenly by each individual quiescent cell, and some cells exit quiescence faster than others. Hence, time generates heterogeneity within quiescent cell populations, heterogeneity that, in turn, increases cell population resilience and robustness to time. In this review, we discuss some of the loops that link quiescence and time.

## 1. Introduction

In the Cambridge Dictionary, quiescence is defined as “the state of being temporary quiet or not active”. In biology, quiescence is a term that embraces many physiological situations. At the cellular level, quiescence is operationally defined as temporary and reversible arrest of proliferation. This cellular state has the greatest prevalence on the planet since most cells—from microbes to cells within complex tissues—spend the majority of their life non-dividing, waiting for signals to reproliferate. In fact, in mammals, quiescent cells are much more numerous than their proliferating counterparts [1]. While research on cellular quiescence was rather confidential until the end of the 20th century, the development of new techniques allowing the study of this amazingly complex cellular state and the recognition of its implication in several human diseases, including cancer [2,3,4], have tremendously stimulated the field. It is now widely acknowledged that quiescence is not simply a passive resting condition but rather an actively initiated and maintained cellular state [5,6,7].

In unicellular organisms, quiescence is most of the time triggered by adverse environmental conditions, such as nutrient exhaustion [8,9,10]. In contrast, in multicellular organisms, cellular quiescence is not associated with scarcity but is rather controlled by complex physicochemical signals emanating from both the niche and the macroenvironment [4,11]. In fact, depending on the environmental input that triggers quiescence entry, quiescent cell properties drastically vary. For example, in budding yeast, the nature of the limiting nutrient significantly impacts cell viability in quiescence [12]. Furthermore, the transcriptome, proteome, and metabolome of quiescent yeast cells differ depending on whether they have ceased to proliferate following nitrogen, carbon, or phosphate starvation [13]. Similarly, human fibroblasts that have entered quiescence through contact inhibition, mitogen withdrawal or loss of adhesion display gene expression profiles that are different from one another [14]. Hence, both the micro- and macro- environment shape quiescence. Consequently, in starved unicellular organisms, quiescence is most of the time associated with low metabolic activity. In comparison, in multicellular organisms, quiescent cells are not necessarily “inactive”, some of them being even more “active” metabolically than their proliferating counterparts [15]. In this case, the metabolic “activity” of quiescent cells depends on their nutritional status and the biological functions that cells have to perform. Thus, both the input (the reason of entering quiescence) and the output (the cellular consequences of entering quiescence) are highly diverse depending on the cell type.

The signalling pathways, including the transcription factors, involved in the quiescence establishment and maintenance, and the quiescent cell’s metabolism properties have been recently described elsewhere and will not be discussed here [3,4,6,15]. In this review, our aim is to question the complex relationships between cellular quiescence and time. We will refer to time as a chronological concept, although time can be envisioned differently [16], and refer to ageing as the biological processes impacted by the effect of time [17,18,19]. We will try to untangle how time influences individual quiescent cell properties and quiescent population robustness, and reciprocally, how quiescence flexibility and heterogeneity play a pivotal role in facing ageing.

## 2. The Multiple Scales of Ageing

The effects of time on biological life operate at multiple scales. The extent to which individual cell ageing is connected to and responsible for ageing at the whole organism level is a complex question. As soon as in the mid-1950s, D. Harman proposed that ageing is caused by the harmful effects of free radicals that build up during life. He thus introduced the notion of a direct and linear relationship between time and ageing [20]. The following 60 years of studies have shown that in multicellular organisms, many processes are at work during ageing. The dynamic field of geroscience is now focusing on understanding how the molecular mechanisms of ageing contribute to age-related diseases [21,22]. In 2013, ten hallmarks of ageing have been posited: genomic instability, telomere attrition, epigenetic alteration, loss of proteostasis, deregulation of nutrient sensing, mitochondrial dysfunction, cellular senescence, autophagy, stem cell exhaustion and alteration of intercellular communications [23]. Some of these hallmarks concern intracellular molecular processes, others rather concern tissue homeostasis, but all support the idea of ageing as a continuous and unidirectional process of decline.

In humans, it is now widely recognized that the occurrence of ageing-related diseases does not follow the linear trend of a proportional increase with age but rather accelerates at specific points throughout the lifespan. At the molecular level, many studies have reported the nonlinear alterations of macromolecules with the age of the individual. This has been particularly well documented for DNA methylation. Indeed, it was shown that this epigenetic mark exhibited nonlinear changes in intensity with age at DNA specific sites, some accelerating for the young and others slowing down for the elderly [24], with sudden hypermethylation events being observed at specific stages of life [25]. Recent deep multi-omics profiling on a longitudinal human cohort has shown that many molecular markers exhibited a nonlinear pattern throughout the ageing process [26], suggesting that ageing occurs in discrete steps rather than as a continuous process. More and more comprehensive multi-omics approaches have explored molecular changes that occur during ageing, and, helped by artificial intelligence, highly complex combinatory methods have allowed the building of “ageing machines” that are powerful enough to consider many molecular parameters, even those that vary nonlinearly with time [27].

The development of single cell multi-omics approaches with an extraordinary precision has opened up further new avenues for exploring the ageing process at the cellular level. Indeed, a combination of mass spectrometry, DNA sequencing, transcriptomic, epigenetic marks, and 3D genome organization analyses are now performed at the single cell level. Combined with high resolution imaging-based methods, these multi-omics approaches can take into account both space and time [28]. All these incredibly complex multimodal analyses converge to demonstrate that within a single body, organs do not age at the same rate [29,30] and that within the same organ, cell type composition and proportion vary with time [31]. These works revealed significant cell type-specific variations that extend far beyond differences in turnover rates and thus demonstrate that not all cell types age at the same speed. This is exemplified in the brain, where different types of neurons exhibit substantial variation in age-related changes in gene expression [32]. In an amazing study, Lu and colleagues have followed all these parameters and have launched an ageing atlas of *Drosophila*. By studying a single-nucleus transcriptomic, they tracked more than 150 distinct cell types and analyzed changes in tissues of male and female flies across their life span and as such portrayed ageing at the single cell level in a full organism [33]. Together, these studies illuminate the nonlinearity of ageing at the organism, organ, tissue, and cell type scales.

## 3. Time Generates Heterogeneity

What could be the cause of this nonlinearity? Could this be the result of heterogeneity in the properties of individual cells in facing age? To tackle these questions, a valuable model is quiescent cell populations. Yet, even when we narrow the focus to clonal cells, these populations are not as homogeneous as they appear, and time itself is generating heterogeneity.

In clonal quiescent cell populations, all cells may not have the same history. In 1961, Hayflick and Moorhead showed that human fetal fibroblasts in culture do not divide indefinitely but undergo a limited number of divisions, then enter senescence [34]. This phenomenon called “replicative ageing” has now been documented from yeast to human. It is acknowledged that a major causal feature of replicative senescence is telomere erosion, a process in which the telomeres lose 50–200 base pairs at each division, due to the inability of DNA polymerase to completely replicate the 3′ end of the DNA strand. In addition, dividing cells accumulate epigenetic marks, DNA mutations, and other damaged macromolecules. Consequently, at the individual cell level, replicative ageing was theorized as linear. Multi-omics studies have indeed recently demonstrated that senescence entry is a gradual process [35]. Whether replicative ageing is relevant for ageing at the whole-body level remains controversial [36], but the cumulative nature of the replicative age does not call into question as the two phenomena are conceptually linked in cellular models in which cell division is symmetric and conservative. However, when cells divide asymmetrically, the linearity may be broken as new daughter cells can be born “young”, the “mother” cell keeping all the damages. This process called rejuvenation, somehow resets time and confers to the population a non-uniform ageing [37,38]. Therefore, replicative age, while cumulative, can possibly generate quiescent cells with different history and as such, different individual properties.

Another layer of heterogeneity comes from the fact that cells do not enter quiescence from the same stage in the cell cycle. The concept that posits that quiescent cells are all arrested at the same point in G1 (the restriction point), proposed by Pardee’s was widely spread by the textbooks [39]. However, it is now demonstrated that cells can be quiescent not only in G1, but also in other cell cycle phases depending on the species, the cell type, or the quiescence input signal. For example, *Saccharomyces cerevisiae* cells enter quiescence mainly from G1, but they can accidentally arrest in other cell cycle phases [40]. Furthermore, *Schizosaccharomyces pombe*, *Cryptococcus neoformans*, *Tetrahymena pyriformis*, *Zea mays* shoot and root meristem cells, *Xenopus* neural progenitors, and many mammalian cell types, such as neural stem cells, can enter quiescence in G1 or G2 depending on external stimuli [41,42,43,44,45,46]. Furthermore, in mammals, oocytes are arrested in the diplotene stage of prophase I of meiosis. This arrest occurs during fetal development and persists until puberty [47]. In addition, for mammalian cell types entering quiescence from G1, it was proposed that cells do not halt synchronously at the restriction point but rather form a cohort of cells arrested as a continuum throughout G1 [41,48,49,50]. These asynchronous cell cycle arrests, thus, participate in the non-uniformity of quiescent cell populations.

Regardless of their history, quiescent cells must face directly the harmful effects of time, a process referred to as “chronological ageing”. Indeed, with time, quiescent cells stochastically build up various metabolic by-products and wastes. As they cannot be diluted via cell division, these compounds have the potential to damage macromolecules and, thus, cellular machineries [51]. For example, quiescent cells may accumulate DNA mutations and/or epigenetic marks that could modify gene expression and hence protein quantity and/or quality. Proteins can also be directly damaged with a panel of so-called “degenerative protein modifications”, such as oxidation, carbonylation, deamidation, etc., that may alter protein structure and functions [52]. Lipids are also age-targets, and their alteration may not only have direct impacts on membrane composition but can also produce by-products that are harmful for other cellular components [53,54,55]. Stochastic macromolecule alteration is obviously greatly influenced by the environment, since harsh conditions such as UV or toxic compound exposure somehow accelerate the deleterious effect of time. The niche also plays a crucial role, as extracellular matrix biophysical parameters and signals secreted by neighbouring lineages can influence the ageing process [11,56,57]. Thus, with time, stochastic macromolecule alterations shape cell individual experience and are another source of heterogeneity within clonal quiescent cell populations.

Overall, the diversity of individual cell properties is shaped by a combination of both deterministic factors relying on cell history and cumulative stochastic events linked to time and the environment. In fact, studying the appearance of heterogeneity within clonal cell populations through in silico approaches predicts that extrinsic factors (macro- and micro-environments) generate distinct subgroups in which cells share common features, while intrinsic factors (stochastic mutations, epigenetic cues, and macromolecule damages) create a broad spectrum of cells with different individual properties [58]. Obviously, quiescent population heterogeneity evolves with time, as described in muscle stem cells [59], illustrating that population heterogeneity itself is dynamic (Figure 1).

## 4. Heterogeneity and Robustness in Facing Time

As exemplified above, time influences the individual quiescent cell’s characteristics, thus participating in the generation and the complexification of heterogeneous cell populations. But does this heterogeneity translate into different individual behaviour in facing time?

In *Drosophila*, cell cycle arrest heterogeneity directly influences the cells’ ability to exit quiescence, as G2-arrested neural stem cells tend to reactivate more rapidly than those arrested in G1 [41]. In fact, quiescence entry in G2 has been associated with efficient stem cell regeneration in a variety of organisms, such as zebrafish, hydra, and axolotl [59]. Thus, the cell cycle stage at which cells enter quiescence does influence quiescent cell properties.

In *S. cerevisiae*, quiescent cell populations are tremendously heterogenous, this heterogeneity evolving with time. In this species, carbon exhaustion causes proliferation cessation that goes with a multitude of cellular reorganizations. For example, cells entirely reshape their nucleus, assembling a stable microtubule bundle form their centrosome, compacting their DNA and clustering their telomeres [60,61,62,63,64]. Cells may also assemble various cytoplasmic foci in which specific enzymes are embedded in inactive forms, such as RNA processing enzyme containing-P-bodies [65], or proteasome storage granules [66]. Some of these cellular reorganizations do form in the majority of the cells regardless of their history, while other are rather rare. This complex combination of reorganizations leads to a population made up of individuals with highly different phenotypes [67]. Importantly, some reorganizations, such as microtubule stabilization, are required for quiescent cell survival upon chronological ageing [68]. The underlying molecular reasons are still unknown, but this phenomenon has been documented in several species [69]. Interestingly, some reorganization are predictors of cell fate. For example, cells that will be capable of surviving in quiescence for weeks reorganize their mitochondria into small cortical vesicles that will reform a dynamic tubular network within few minutes upon quiescence exit. By contrast, quiescent cells that will enter senescence within a couple of days display globular mitochondria that progressively lose their oxphos abilities [70]. Although the interdependence between each cellular remodeling and how they are modulated by time and environmental factors remain mostly unclear, the ability of yeast cells to maintain quiescence and withstand ageing in a given environment is underpinned by their individual cell phenotype (Figure 1).

At the population level, this diversity of phenotypes could be crucial for the survival of the species. Indeed, some quiescent cells with given properties may be more capable of surviving in a cognate environment, while others may be more resilient in other kinds of harsh conditions. The most famous example is persisters. Within a genetically homogeneous population of quiescent bacterial cells, persister cells can survive to environmental stressors such as massive antibiotic exposure. When the stress is removed, persisters generate offspring that are susceptible to the same stress, demonstrating that “persistence” is not genetically encoded [71]. Intriguingly, in several species like *Escherichia coli*, *Salmonella typhimurium*, or *Shigella flexneri*, persister cells assembled a specific subcellular structure called regrowth-delay body, made of multiple proteins such as FtsZ (the tubulin homologue) and FtsA (the actin homologue) that favour re-proliferation [72]. As such, bacterial populations’ heterogeneity can provide selective advantages in changing environments [73]. This is reminiscent of what is known for T-cell populations. Indeed, after activation by an infectious agent, few activated T-cells remain quiescent, creating an “immune memory” that provides a rapid response in case of an eventual re-infection by the same pathogenic agent. This collection of quiescent T-cells with different properties laid the foundation of an effective immune response in changing environments [74]. Interestingly, Muscle Stem Cells (MuSC) in mice display a functional heterogeneity in their responses to environmental stress, such as pollutant exposure, that depends on the level of expression of the transcription factor Pax3 [75]. Thus, heterogeneity in quiescent cell phenotype confers robustness to the population when external conditions are non-optimal or fluctuating.

## 5. Quiescence Deepening

Quiescent cells have not only to survive the deleterious effect of age, but with time, they also have to keep their capacity to reproliferate. In the 1960s, Nancy Bucher was focusing on liver regeneration. She observed in both rats and mice, that “young animals restore tissue mass and cell populations more rapidly than adults” and that “DNA synthesis gets underway more slowly as age increases” [76]. In 1974, Augenlicht and Baserga reported that the longer WI-38 fibroblasts spent in quiescence, the longer was the pre-replicative phase after re-stimulation. They proposed that “G0 is apparently composed of several different states, since the cells go ‘deeper’ into G0 with time”, thereby introducing the notion of quiescence deepening [77]. As early as in 1976, Baserga proposed that “prolonged quiescence may bear some resemblance to the process of aging.” Then, studies from the Baserga, the Soprano, and the O’Farrell laboratories repeatedly reported that the longer the time in quiescence, the longer it takes for quiescent cells to return into the proliferation cycle [78,79,80,81]. More recently, similar observations were made in *E. coli* [82], in budding [83] and fission yeast [84], in *C. elegans* [85] and in many mammalian models, including rat embryonic fibroblasts [86], hematopoietic stem cells, neural stem cells, and muscle stem cells [87,88,89,90].

Another characteristic of quiescence deepening is the “desensitization” to the re-proliferation stimulus. For example, with increasingly longer serum starvation, rat embryonic fibroblasts require a greater serum stimulation strength (concentration) to re-enter the cell cycle than cells in shallower quiescence [86,89]. This is reminiscent of the primary observation reporting that short-term quiescent WI-38 cells bind substantially higher amounts of EGF [91] and internalize EGF receptor complexes more rapidly [92] than long-term quiescent cells. Therefore, in several mammalian models, quiescence deepening is characterized by both a reduced sensitivity to stimulation and slower re-entry into the cell cycle.

The molecular basis of the quiescence deepening process is only beginning to be understood. It has long been observed that both the transcription and translation rates decrease and that the chromatin condensation increases with time in quiescence [9,80,93]. It was also shown that gene expression and metabolic profiles evolved with time in quiescence, as exemplified by studies in fibroblasts [14,89]. Hence, a gene signature, derived from transcriptomic approaches and called the quiescence depth score, has been proposed to predict the deepness of quiescence in both human fibroblast and mouse neural stem cells [94]. While we can speculate that the molecular processes underlying quiescence deepening likely involve a complex interplay between various molecular pathways, converging themes start to emerge. In fibroblasts, neural and muscle stem cells, decreased lysosomal autophagy drives progressively cells into deeper quiescence, likely by controlling the level of protein aggregates [89,95,96,97,98]. Furthermore, quiescence depth is influenced by the Rb-E2F network that acts as a bi-stable switch. Through experiments in fibroblasts and computational modeling, it was shown that different components of the Rb-E2F network can be manipulated to fine-tune quiescence depth. In fact, deep quiescent cells have a higher activation threshold of the Rb-E2F switch, requiring stronger stimulation to re-enter the cell cycle [86,93,97]. In addition, circadian proteins, such as cryptochrome (Cry) and Rev-erb, also regulate quiescence depth in rat embryonic fibroblasts, by downregulating cyclin D/Cdk4,6 activity, leading to an increased need for Rb-E2F activation threshold [99]. As with circadian proteins, the Rb–E2F switch is connected to many other regulatory pathways, including Notch, Hes1, p21, etc., all being involved in quiescence establishment. Hence, it was proposed that the Rb–E2F switch serves as a hub that integrates signals that quantitatively tune quiescence depth [93]. The bone morphogenetic protein (BMP) also plays a crucial role in regulating the depth of quiescence, notably in adult neural stem cells, where it was shown that BMP4 in combination with FGF2 induces a shallow quiescent state, whereas BMP4 alone induces a deeper quiescent state [100,101]. The implication of BMP in quiescence has recently been discussed in [102]. Finally, recent deep learning approaches leveraging both metabolic and epigenetic data have identified small molecules and genetic modulators that accurately predict quiescence depth in diverse biological contexts [103].

Taken together, all these studies converge on the idea that quiescence is not a uniform state but exists as a continuum that deepens with time. In some models, this trajectory may end in senescence, as evidenced by the significant similarity between the transcriptomic profiles of deep quiescent cells and senescent cells [104]. This idea has been elegantly discussed in [93]. Yet, our understanding of the molecular mechanisms that regulate the transitions between proliferation, quiescence, senescence, and cell death still remains rather limited and may be very different depending on the cell type and the organism.

## 6. Quiescence Exit Efficiency

In many stem cell lineages, it has been observed that quiescence exit is not homogenous and, as such, defines subpopulations displaying various swiftness in their ability to re-enter the proliferation cycle [4]. In adult muscles, quiescent MuSCs must respond to environmental signals in order to ensure the regeneration of the tissue. MuSCs quiescence is actively maintained through a combination of intrinsic factors and by the niche, including myofibers and other muscle-resident cells together with the basal lamina and adhesion molecules [105]. MuSCs are highly heterogeneous, and subpopulations can be distinguished by different gene expression profiles or surface markers [106]. MuSCs that display features of “deeper” quiescence display dynamic cellular projections. These structures retract following muscle injury in order to allow MuSC reproliferation, a reorganization occurring through a rapid cytoskeletal remodelling mediated by Rac/Rho GTPase signalling [107]. In fact, MuSCs exist in two different quiescent states: a dormant state that is thought to preserve the long-term regenerative capacity of muscle tissue, and a more responsive, primed state called G(Alert) [108]. MuSCs in the G(Alert) state exhibit an increased cell size, elevated transcriptional and mitochondrial activity, and higher ATP levels compared to their deeply quiescent counterparts [2]. High levels of PAX7 and low levels of MYF5 delineate the deeply quiescent MuSC subpopulation [109]. The transition from deep quiescence to G(Alert) is orchestrated by systemic signals released upon tissue injury that involve the hepatocyte growth factor activator (HGFA) and HMGB1, which activate the mTORC1 and the CXCL12/CXCR4 pathways [108].

Similarly, among hematopoietic stem cell populations (HSCs), long-term HSCs (LT-HSCs) are deeply quiescent and can be “activated” to become short-term HSCs (ST-HSCs), which, while still quiescent, are primed for cell cycle entry [88,110]. This transition is tightly controlled by extrinsic cues from the bone marrow niche and intrinsic regulators, such as the expression of cell cycle regulators like CDK6 or CD38 [111,112]. Since LT-HSCs are characterized by their potent self-renewal capacity over the entire lifespan of an organism, it was proposed that LT-HSCs ensure a reservoir for long-term hematopoietic integrity, while ST-HSC allow the rapid expansion of HSCs in response to the physiological demands or stress [113,114].

Likewise, primed quiescent neural stem cells (p-qNSC) characterized by a high level of LRIG1 and CD9 have been described in adult mice brains [115]. These cells display a reduced glycolytic metabolism, a downregulated Notch pathway, and upregulation of protein synthesis [116,117]. Primed qNSC are poised for rapid cell cycle re-entry and can efficiently engraft into the adult subventricular zone niche, thereby contributing to tissue regeneration and repair. Interestingly, qNSC from young and old mice have no major differences, neither in their levels of DNA damage nor in their transcriptional profile. Yet, young qNSCs transit more efficiently to a primed-quiescent state than older qNSCs, but once activated, both young and old qNSCs display a similar capacity to undergo several rounds of division [118].

In the above examples, quiescence is described as a binary state with “deep/slow” or “primed/fast” states, the latter being induced via an extrinsic pre-stimulation from the niche. Yet, could quiescence “responsiveness” be more gradual and exit as a continuum within a population? Could intrinsic individual cell features be sufficient to support a variation in quiescence exit efficiency, independently of a pre-stimulation by the niche? In quiescent *S. cerevisiae* population, a continuum in cell size distribution correlates with a continuum in quiescence exit efficacy, the tendency being that the bigger the cell, the more prone it is to exit quiescence efficiently. Yet this is not an absolute rule, and while daughter cell quiescence exit efficiency is mostly controlled by a sizer, in mother cells, it is neither influenced by the cell volume nor by the cell replicative age, but rather depends on a “timer” [83]. In quiescent rat embryonic fibroblast populations, some quiescent cells are capable of responding to a short serum stimulation, while others need to be in contact with the stimulus for a longer period of time to be able to exit quiescence [119]. Thus, a continuum of quiescent cell responsiveness does exist in a clonal cell population in which cells have experienced the same environmental history [119]. This heterogeneity in quiescence exit efficiency seems to be influenced by a “memory” combining deterministic factors, such as the cell size at the time of quiescence induction, with stochastic elements, such as noise in the Rb-E2F bi-stable switch [119].

Altogether, these observations underscore that heterogeneity in quiescence exit efficiency is a common feature of quiescent populations of many cell types. Depending on the context, this heterogeneity may be the consequence of stochastic individual features or can be induced by deterministic events that confer cells a predictable behaviour. This diversity may enhance the robustness and the resilience of population facing age and/or environmental fluctuations. In tissues, this may safeguard against acute depletion of the quiescent cell reservoir in response to a single activating stimulus. In single cell species, such as bacteria, such intrapopulation heterogeneity has been proposed to provide a bet-hedging advantage, ensuring that not all cells respond identically to reactivation signals and resume growth simultaneously. This strategy may protect the population from sudden environmental threats and support long-term survival [120]. More generally, heterogeneity in quiescence exit efficiency could enhance the long-term survival and adaptability of the population in fluctuating environments.

## 7. Conclusions and Model

While it is experimentally challenging to distinguish quiescence deepening and priming—as deepening is synonymous with delayed quiescence exit—these two processes are not conceptually equivalent. By definition, quiescence is reversible and, as such, is an “aller-retour”. The “aller“ part of the ticket corresponds to the length of time a cell has spent in quiescence. This time is incompressible. During that trip, cells accumulate damages, not necessarily with a linear trend, but that impact quiescent cell individual properties and, as such, may contribute to the rising of population heterogeneity. The “retour” is the time a cell needs to exit quiescence id est to re-enter the proliferation cycle. The length of the “retour” trip depends on the time spent in quiescence, the environment and the cell individual features (Figure 2A). The “retour” is related to the “aller” in that the longer a cell has been in quiescence, the more time it needs to return into the proliferation cycle. But for a similar “aller” time, the time needed for the retour differs depending on the quiescence depth (Figure 2B(a)), the cell capacity to sense and react to the quiescence exit signal (Figure 2B(b)), and the speed at which the cell will exit quiescence (Figure 2B(c)). Together, these models highlight the multifaceted nature of quiescence and its relationships to time, demonstrating that a quiescent cell population can encompass a collection of cellular universes that co-exist at the same time.

## Figures and Tables

**Figure 1 biomolecules-15-00960-f001:**
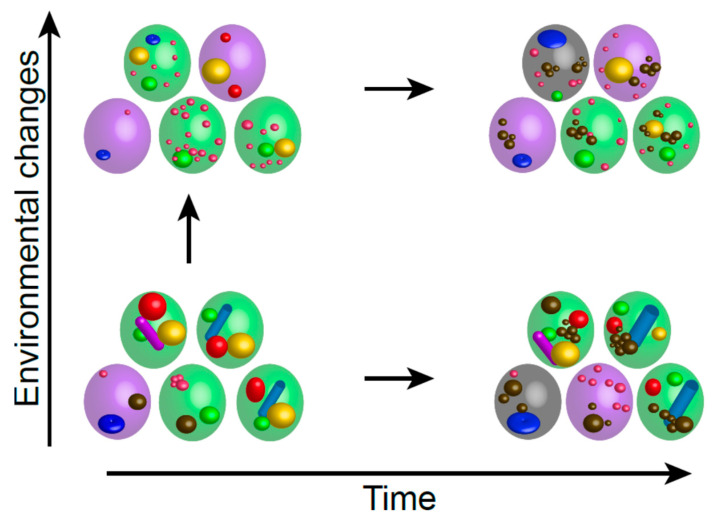
Each cell displays unique features, such as the organization of specific cellular machineries or cellular damages—as schematized by the different objects drawn inside the cells—thereby conferring to each cellspecific individual properties. These individual cell features evolve with time and/or with the environmental fluctuations, hence impacting cell fate (quiescent cells are green, senescent cells are violet, and dead cells are grey).

**Figure 2 biomolecules-15-00960-f002:**
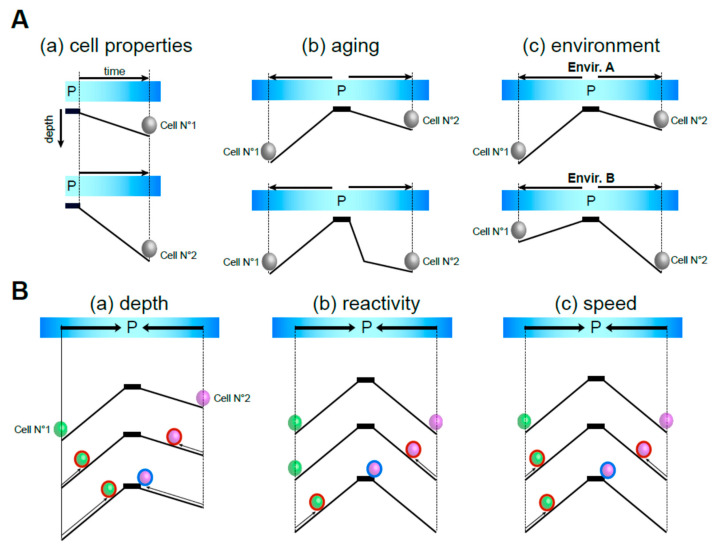
(**A**) Depending on their individual properties, time, and environment, cells deepen into quiescence with different slopes. (P stands for proliferation). (**B**) The quiescence exit process is unique to each individual cell and depends on the depth of quiescence, the quiescence exit speed, and the reactivity of quiescent cells to the quiescence exit signal. Schemes represent two individual cells with (a) the same reactivity, the same quiescence exit speed but different quiescence depth; or (b) same reactivity, same quiescence depth but different quiescence exit speed; or (c) same quiescence depth, the same quiescence exit speed but different reactivity to the quiescence exit signal. Quiescent cells that are “activated for exit” are circled in red and cells entering into the proliferation cycle are circled in blue.

## Data Availability

No new data were created or analyzed in this study.

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
