# Peer review of "Quiescence Multiverse"

_biomolecules, 2025, doi:10.3390/biom15070960_

Round 1

Reviewer 1 Report

Comments and Suggestions for Authors

Laporte and Sagot wrote an interesting review on the nature of quiescence in the context of time. They discuss aspects of heterogeneity in the quiescent state and how that heterogeneity could impact cell function.

Overall comments:

The review nicely takes examples from highly diverse cell models, and highlights throughout which models a finding was made in. What could help make the text accessible to a wider audience is to provide a table or visual overview of the major quiescence models that are discussed. Are there clear examples of findings that are shared between models and findings that are specific to one or the other model? 

Specific points:

Line 23: please cite the dictionary the quoted definition is from. 

Line 29: true but this is more due to the low numbers of proliferating cells than the abundance of quiescent cells. Most cells in the body are terminally differentiated cells. 

Line 58: please mention the other concepts of time that are not chronological in nature.

Line 62: please delete the instructions.

Line 85: Time has a single direction, but as is noted later on, cells can rejuvenate, either artificially (i.e. induced pluripotency) or biologically (i.e. fertilized oocytes, or asymmetric division plus non-random segregation of damaged goods). This would argue against the unidirectionality of aging as a biological process.

Line 113: have launched

Line 144: comes

Line 153: should mention the female germ line where cells arrest in first meiosis profase.

Line 155: one could argue that it would be highly inefficient for a cell to complete G1 only to stop at the restriction point. This could certainly occur in single cell organisms or cell culture systems, but it seems less likely in the context of multicellular organisms, which provide much stricter space and nutrient limitations on the cells. Later in the review, the authors briefly remark on about the size of yeast of and alert stem cells. This could perhaps be expanded upon.

Line 163: why would “toxic compounds” not be eliminated by quiescent cells? Please cite studies showing that. While any activity to repair or remove will likely be toned down in the quiescent state, the same would be the case for any activity that results in “toxic compounds”. For example, quiescent stem cells have much lower mitochondrial content, and reside in hypoxic environments, which would likely result in a lower burden of oxygen radicals derived from respiration. Certain repair mechanisms, such as homologous DNA repair mechanisms, will be unavailable until the cell enters S-phase, but are quite effective then. Could there be repair mechanisms specific to the quiescent state?

Line 188: what are all the objects drawn inside the cells? Please provide a legend and a more detailed caption.

Line 193: the quiescent cell’s characteristics

Line 218: what does senescence have to do with quiescence? Senescent cells by definition are not quiescent.

Line 227: observations similar to persistence exist in mammalian contexts where subsets of quiescent stem cells can resist environmental challenges (i.e. der Vartarian et al cell stem cell 2019). Unlike in the bacteria example cited, a complicating factor here is that positional information may play a role in determining the quiescence heterogeneity. Perhaps this is something that could be explored further? For example, in mice, stem cell quiescence varies depending on the location in the body. Neural stem cells (dentate gyrus vs subventricular zone (i.e. Fontán-Lozano et al frontiers in oncology 2020)), muscle stem cells (diaphragm vs hind limb muscles (i.e. der Vartarian 2019)), and hematopoietic stem cells (spine vs limb bones (i.e. Verovskaya et al journal of experimental medicine, 2014)) all show heterogeneity in the quiescent state depending on where in the body the niche is located. Similarly, in contact inhibition models, it is the cells that are contact inhibited that enter quiescence, while proliferation is more likely to continue in sparser populated areas of the same culture dish. In other words, to what extent does positional information impact the heterogeneity of the quiescent state? 

Line 259: muscle stem cells. The sentence reads odd given the placement of the commas. I would suggest adding “stem cells” after “hematopoietic” and “neural”.

Line 263: this paragraph is about quiescence depth, not aging, and accordingly it should cite a study of quiescence depth at the same age. The cited study shows a reduced response of aged cells, which is not the same as quiescence depth, which has been well documented in young animals.

Line 295: given the definition of quiescence as a reversible exit from the cell cycle, it is most certainly an on/off-state, albeit one with multiple, often exclusive and irreversible, off-states (proliferation, differentiation, senescence, apoptosis, etc). One could perhaps argue that the on-state is more of a dimmer switch (of quiescence depth) that can burn more or less bright.

Line 300: as above, there are other options for a quiescent cell that deserve to be mentioned, most certainly proliferation.

Line 351: what is a “sizer”?

Line 382: needs to

Comments on the Quality of English Language

The article is overall well written and easy to follow. It would however benefit from a grammar check. I highlighted a few sentences.

Author Response

Reviewer N°1

The review nicely takes examples from highly diverse cell models, and highlights throughout which models a finding was made in. What could help make the text accessible to a wider audience is to provide a table or visual overview of the major quiescence models that are discussed. Are there clear examples of findings that are shared between models and findings that are specific to one or the other model? 

Our review is not exhaustive and many other examples could be found in the literature. Creating a table/figure would be very reductionist. We preferred putting together some models (fig1 and Fig2) that can be testable in all cell types. 

Specific points:

Line 23: please cite the dictionary the quoted definition is from. See line 23

Line 29: true but this is more due to the low numbers of proliferating cells than the abundance of quiescent cells. Most cells in the body are terminally differentiated cells. 

We have tried to outline the conceptual differences between quiescence and differentiation in a previous review (Laporte D, Sagot I. Physiology. 2025 doi: 10.1152/physiol.00036.2024.). This concept was also discussed widely in several others reviews such as (Stem Cell Quiescence: Dynamism, Restraint, and Cellular Idling, 2020, C. van Velthoven, T. Rando).

Line 58: please mention the other concepts of time that are not chronological in nature. See line 60.

Line 62: please delete the instructions. Done

Line 85: Time has a single direction, but as is noted later on, cells can rejuvenate, either artificially (i.e. induced pluripotency) or biologically (i.e. fertilized oocytes, or asymmetric division plus non-random segregation of damaged goods). This would argue against the unidirectionality of aging as a biological process. Indeed. Please refer to line 134.

Line 113: have launched. This has been fixed

Line 144: comes. This has been fixed

Line 153: should mention the female germ line where cells arrest in first meiosis prophase. One sentence has been added line 148-149.

Line 155: one could argue that it would be highly inefficient for a cell to complete G1 only to stop at the restriction point. This could certainly occur in single cell organisms or cell culture systems, but it seems less likely in the context of multicellular organisms, which provide much stricter space and nutrient limitations on the cells. Later in the review, the authors briefly remark on about the size of yeast of and alert stem cells. This could perhaps be expanded upon.

We agree with the reviewer. Cell cycle arrest upon quiescence establishment is a fascinating issue that imply a description of all the signaling pathways involved in this process and their vast diversity depending on model organisms. In our opinion, this topic deserves a dedicated review. That is why we chose to keep this point rather short here.

Line 163: why would “toxic compounds” not be eliminated by quiescent cells? Please cite studies showing that. While any activity to repair or remove will likely be toned down in the quiescent state, the same would be the case for any activity that results in “toxic compounds”. For example, quiescent stem cells have much lower mitochondrial content, and reside in hypoxic environments, which would likely result in a lower burden of oxygen radicals derived from respiration. Certain repair mechanisms, such as homologous DNA repair mechanisms, will be unavailable until the cell enters S-phase, but are quite effective then. Could there be repair mechanisms specific to the quiescent state?

We have shortened this part (see lines 156-57). The process of toxic compound elimination in quiescent cells, and more generally the activity of degradation machineries such as the proteasome, or autophagy together with the activity of repair machineries in quiescence vary drastically depending on the cell type, its age and its environment. This is not the focus of our review and we believe this very interesting point deserves a full new review.

Line 188: what are all the objects drawn inside the cells? Please provide a legend and a more detailed caption. We have implemented the figure 1 legend according to the reviewer’s comment.

Line 193: the quiescent cell’s characteristics. This has been fixed.

Line 218: what does senescence have to do with quiescence? Senescent cells by definition are not quiescent. Indeed. This was just for the comparison.

Line 227: observations similar to persistence exist in mammalian contexts where subsets of quiescent stem cells can resist environmental challenges (i.e. der Vartarian et al cell stem cell 2019). Thank you for pointing at this work. We have added a sentence (line 234-37).

Unlike in the bacteria example cited, a complicating factor here is that positional information may play a role in determining the quiescence heterogeneity. Perhaps this is something that could be explored further? For example, in mice, stem cell quiescence varies depending on the location in the body. Neural stem cells (dentate gyrus vs subventricular zone (i.e. Fontán-Lozano et al frontiers in oncology 2020)), muscle stem cells (diaphragm vs hind limb muscles (i.e. der Vartarian 2019)), and hematopoietic stem cells (spine vs limb bones (i.e. Verovskaya et al journal of experimental medicine, 2014)) all show heterogeneity in the quiescent state depending on where in the body the niche is located. Similarly, in contact inhibition models, it is the cells that are contact inhibited that enter quiescence, while proliferation is more likely to continue in sparser populated areas of the same culture dish. In other words, to what extent does positional information impact the heterogeneity of the quiescent state? 

We totally agree that the niche/location in the body is another source of quiescent cell heterogeneity. In one preliminary version of this review, we dedicated a paragraph to that idea, but in fact, the literature is so rich that this area of research deserves a full review. Further, the review by Fiore et al, Front Cell Dev Biol does a superb job in tackling the relationships between quiescence and the niche. For those reasons, we chose to remove that part and to focus on the relationship with quiescence and time – not space.

Line 259: muscle stem cells. The sentence reads odd given the placement of the commas. I would suggest adding “stem cells” after “hematopoietic” and “neural”. This has been fixed.

Line 263: this paragraph is about quiescence depth, not aging, and accordingly it should cite a study of quiescence depth at the same age. The cited study shows a reduced response of aged cells, which is not the same as quiescence depth, which has been well documented in young animals.

The study by Kwon and colleagues (Cell reports, 2017) shows that “deep quiescent cells required stronger growth stimulation to re-enter the cell cycle and initiate DNA synthesis compared to shallow quiescent cells.” They compared the ability of rat embryonic fibroblasts (REF/E23 cells) that have been in quiescence during an increasing amount of time after contact inhibition and demonstrated that the longer they have been contact inhibited, the more serum they need to exit quiescence: “serum concentrations required for 4D-STA [serum starvation] deep quiescent cells were roughly twice those required for 2D-STA shallow quiescent cells (Figures 1E and 1F).”

Similarly, the study by Fujimaki et al (PNAS, 2019) demonstrates that “With increasingly longer serum starvation, cells moved into deeper quiescence, requiring a longer time and a greater serum stimulation strength (concentration) to reenter the cell cycle than cells in shallower quiescence. For example, when the duration of serum starvation increased gradually from 2 to 14 d, the cell cycle reentry time upon 20% serum stimulation increased from <19 h to ∼25 h (to achieve ∼60% EdU+ rate) (Fig. 1A and SI Appendix, Fig. S1C), and the serum concentration required for cell cycle reentry increased from <2% to ∼8% (to achieve ∼60% EdU+ measured at the 25th hour of serum stimulation) (Fig. 1B).

We thus believe that it exemplified that cell in deeper quiescence are less “sensitive to serum” than shallow quiescent cells. But we have modified our sentence to make it more clear.

Line 295: given the definition of quiescence as a reversible exit from the cell cycle, it is most certainly an on/off-state, albeit one with multiple, often exclusive and irreversible, off-states (proliferation, differentiation, senescence, apoptosis, etc). One could perhaps argue that the on-state is more of a dimmer switch (of quiescence depth) that can burn more or less bright.

Indeed, this is exactly what we meant by “quiescence is not an on/off state”. But we have changed it to a “is not a uniform state”.

Line 300: as above, there are other options for a quiescent cell that deserve to be mentioned, most certainly proliferation. We now mention proliferation.

Line 351: what is a “sizer”?

In several unicellular models, cell cycle re-entry is controlled by the size of the cell – [sizer]. This was shown for Clamydomonas (Cell cycle control by timer and sizer in Chlamydomonas. Donnan L, John PC. Nature. 1983 Aug 18-24;304(5927):630-3. doi: 10.1038/304630a0.), for fission yeast (The size control of fission yeast revisited. Sveiczer A, Novak B, Mitchison JM. J Cell Sci. 1996 Dec;109 ( Pt 12):2947-57. doi:10.1242/jcs.109.12.2947.), for budding yeast (The effects of molecular noise and size control on variability in the budding yeast cell cycle. Di Talia S, Skotheim JM, Bean JM, Siggia ED, Cross FR. Nature. 2007 Aug 23;448(7156):947-51. doi: 10.1038/nature06072.). It has been modelized (A mathematical model for cell size control in fission yeast. Li B, Shao B, Yu C, Ouyang Q, Wang H. J Theor Biol. 2010 Jun 7;264(3):771-81. doi: 10.1016/j.jtbi.2010.03.023) and reviewed for example in (Controlling cell size through sizer mechanisms. Facchetti G, Chang F, Howard M. Curr Opin Syst Biol. 2017 Oct;5:86-92. doi: 10.1016/j.coisb.2017.08.010.).

Line 382: needs to. This has been fixed

Reviewer 2 Report

Comments and Suggestions for Authors

The review is well written and presents a good insight into the different stages of quiescence. The review is important and timely because often quiescence is misunderstood as a single, uniform state. The authors mainly omit molecular details in signalling and metabolic pathways in the the initial 2/3 of the reviews. Yet dive into molecular depths into 'Quiescence Exit Efficiency'. I am puzzled that BMP signalling which has been shown to be crucial for the entry into deep quiescence of many stem cells has not been mentioned. A paragraph about the role of BMP should be added. Figure 2 is not easy to comprehend but I think it could be improved by changing the colour of the cells. In A the pink cell is always the cell in deeper quiescence but the same pink cell in B is now in shallow quiescence and always faster to reactivate. Change pink to green to make the model consistent with the experimental observations. 

A few minor errors:

  1. Although the last paragraph in the introduction is a funny I think it should be deleted.
  2. Drosophila should be always starting with a capital and be in italics.
  3. line 297: by should be replaced with in
  4. line 238- 241: The author mentions T-cells and follow up with a conclusion referring to unicellular organisms which have no adaptive immune system. Delete the unicellular organisms.
  5. line 150: form should be from
  6. line 113: launch should be launched
  7. line 27: form should be from

Author Response

Reviewer N°2

 The review is well written and presents a good insight into the different stages of quiescence. The review is important and timely because often quiescence is misunderstood as a single, uniform state.

The authors mainly omit molecular details in signalling and metabolic pathways in the the initial 2/3 of the reviews. We have added a sentence line 56-57.

Yet dive into molecular depths into 'Quiescence Exit Efficiency'. I am puzzled that BMP signalling which has been shown to be crucial for the entry into deep quiescence of many stem cells has not been mentioned. A paragraph about the role of BMP should be added.

A paragraph was added line 288.

Figure 2 is not easy to comprehend but I think it could be improved by changing the colour of the cells. In A the pink cell is always the cell in deeper quiescence but the same pink cell in B is now in shallow quiescence and always faster to reactivate. Change pink to green to make the model consistent with the experimental observations. Colors have been changed.

A few minor errors:

  1. Although the last paragraph in the introduction is a funny I think it should be deleted.

This was a mistake – it has been removed

  1. Drosophila should be always starting with a capital and be in italics. This has been fixed
  2. line 297: by should be replaced with in. This has been fixed
  3. line 238- 241: The author mentions T-cells and follow up with a conclusion referring to unicellular organisms which have no adaptive immune system. Delete the unicellular organisms. This has been fixed
  4. line 150: form should be from. This has been fixed
  5. line 113: launch should be launched. This has been fixed
  6. line 27: form should be from. This has been fixed